# MCPVerse: An Expansive, Real-World Benchmark for Agentic Tool Use

## Abstract

Large Language Models (LLMs) are evolving from text generators into reasoning agents. This transition makes their ability to use external tools a critical capability. However, evaluating this skill presents a significant challenge. Existing benchmarks are often limited by their reliance on synthetic tools and severely constrained action spaces. To address these limitations, we introduce MCPVerse, an expansive, real-world benchmark for evaluating agentic tool use. MCPVerse integrates more than 550 real-world, executable tools to create an unprecedented action space exceeding 147k tokens, and employs outcome-based evaluation with real-time ground truth for time-sensitive tasks. We benchmarked the state-of-the-art LLMs across three modes (Oracle, Standard, and Max-Scale), revealing that while most models suffer performance degradation when confronted with larger tool sets, the agentic models, such as Claude-4-Sonnet, can effectively leverage expanded tool spaces to improve accuracy. This finding not only exposes the limitations of state-of-the-art models in complex, real-world scenarios but also establishes MCPVerse as a critical benchmark for measuring and advancing agentic tool use capabilities.

## 1 Introduction

The ability of Large Language Models (LLMs) to interact with external tools, typically through function calling, is fundamental to their application in real-world scenarios. This capability allows them to access live data, execute code, and operate other systems, thereby moving beyond their static knowledge. Despite its importance, the evaluation of tool use is hampered by two shortcomings in current benchmarks.

First, many benchmarks rely on artificial tools (Patil et al., 2025; Li et al., 2023b). These often simulate calculators, simplified weather services, or mock shopping carts, whose data formats and interaction patterns bear little resemblance to production systems. This discrepancy allows models to succeed by recognizing superficial patterns rather than demonstrating the robust planning and coordination skills required for real-world tasks. While some benchmarks (Qin et al., 2024) claim to incorporate a wide range of real-world APIs, they often stop short of actual execution due to the complexities of live integration. Evaluation is thus confined to assessing the correctness of the selected tool name and its parameters, rather than the functional outcome of the interaction.

Second, existing benchmarks severely constrain the action space available to the model during evaluation. Even when a large set of APIs is listed, context length limitations force designers to mount only a small subset, often relying on a retrieval module to select a few dozen relevant options per query (Fei et al., 2025; Li et al., 2023b; Qin et al., 2024). This strategy, while keeping prompts within the token limits of current models, prevents the assessment of a model's ability to navigate a vast and complex solution space.

The integration of external tools with large language models was previously hindered by a fragmented ecosystem, where different platforms relied on custom connectors and proprietary protocols. The Model Context Protocol (MCP) was introduced in 2024 as an open standard to address this challenge. By providing a uniform interface for tool access, MCP has spurred rapid adoption across major AI platforms, leading to the creation of hundreds of diverse MCP servers for applications involving web search, file systems, databases, and specialized APIs.

Table 1: Comparison of tool-use benchmark across key dimensions. Executable tools refers to the number of tools that may be executed during the evaluation process. Max mounted tools refers to the maximum number of tools that can be mounted for a single task.

| Dataset | Curation | MCP Support | Outcome-Based Evaluation | Real-Time Ground Truth | Executable Tools (Total / Real / Simulated) | Max Mounted Tools |
|---|---|---|---|---|---|---|
| MCPVerse (Ours) | Human | ✓ | ✓ | ✓ | 552 / 552 / 0 | 552 |
| BFCL-v3 | Human | ✗ | ✓ | ✗ | 76 / 0 / 76 | 37 |
| ToolBench | Synthetic | ✗ | ✗ | ✗ | 0 / 0 / 0 | 17 |
| $\tau$-bench | Synthetic | ✗ | ✓ | ✗ | 28 / 0 / 28 | 15 |
| API-Bank | Human | ✗ | ✓ | ✗ | 73 / 0 / 73 | 8 |
| ToolSandBox | Human | ✗ | ✓ | ✗ | 34 / 0 / 34 | 34 |
| MCPBench | Human | ✓ | ✓ | ✗ | 27 / 27 / 0 | 10 |
| MCP-RADAR | Human | ✓ | ✓ | ✗ | 42 / 42 / 0 | 42 |

The widespread adoption of the MCP establishes a new paradigm for tool use. Building on this foundation, we introduce MCPVerse, a novel evaluation framework grounded in the principles of realism and scale. As shown in Table 1, MCPVerse is distinguished from prior work in three critical ways:

- **Realistic Tasks and Real-Time Verification**: All tasks in MCPVerse are constructed using real-world information, such as map data and flight schedules. To handle time-sensitive queries, we developed dynamic scripts that fetch real-time ground truth, ensuring evaluation accuracy.

- **Expansive Action Space**: We have carefully curated a collection of 65 MCPs, encompassing 552 unique tools that cover a diverse range of functionalities. The combined schemas of these tools exceed $147k$ tokens, surpassing the context and tool-mounting limits of many state-of-the-art models and providing an unprecedentedly large tool space for exploration and exploitation.

- **Hybrid Outcome-Based Evaluation**: Recognizing that a single task can have multiple valid solution paths, our evaluation focuses on the final outcome rather than the specific sequence of tools used. We do not penalize models for deviating from a prescribed path. We employ a hybrid evaluation method: an LLM judges semantic consistency between the outcome and the ground truth, while automated evaluation scripts verify state changes for tasks involving file system modifications or other environmental interactions.

Using our evaluation system, we benchmarked 12 leading LLMs. Our findings show that the top-performing model, Claude-4-Sonnet, achieved an success rate of only $44.2$ at max-scale mode, indicating significant room for improvement across the field. Furthermore, our experiments revealed that many state-of-the-art models are not yet equipped to handle the scale of MCPVerse, facing limitations such as a 64K context length (DeepSeek-V3), a 128-tool limit (GPT-4o-20241120), or a 512-tool limit (Gemini-2.5-Pro).

To accommodate these constraints while still probing performance at scale, we designed three evaluation modes: Oracle mode (only per-question minimal MCPs are provided), Standard mode (all potentially relevant MCPs are provided), and Max-Scale Mode (all 65 MCPs are mounted). Unsurprisingly, most models exhibited performance degradation as the number of available MCPs increased. More strikingly, agentic models like Claude-4-Sonnet, Qwen3-235B-2507 and GLM-4.5 achieved a higher score in Standard mode than in the simpler Oracle mode. This counter-intuitive result suggests that a larger tool space enables emergent solution paths and "hacking" behaviors, allowing capable models to uncover alternatives that are inaccessible under constrained settings.

In summary, our contributions are:

- We introduce MCPVerse benchmark, a comprehensive benchmark built on a large-scale set of executable, real-world tools. The benchmark features meticulously designed tasks, each annotated with ground truth (or an evaluation script for time-sensitive queries), as well as additional metadata such as required MCPs, complexity level, and task type.

- We build an end-to-end, automated evaluation system that facilitates multi-step interactions between the LLM agent and MCP tools. The system assesses the final outcome using our hybrid, outcome-based metric, which combines an LLM-as-a-judge for textual answers with dedicated scripts for verifying environmental state changes.

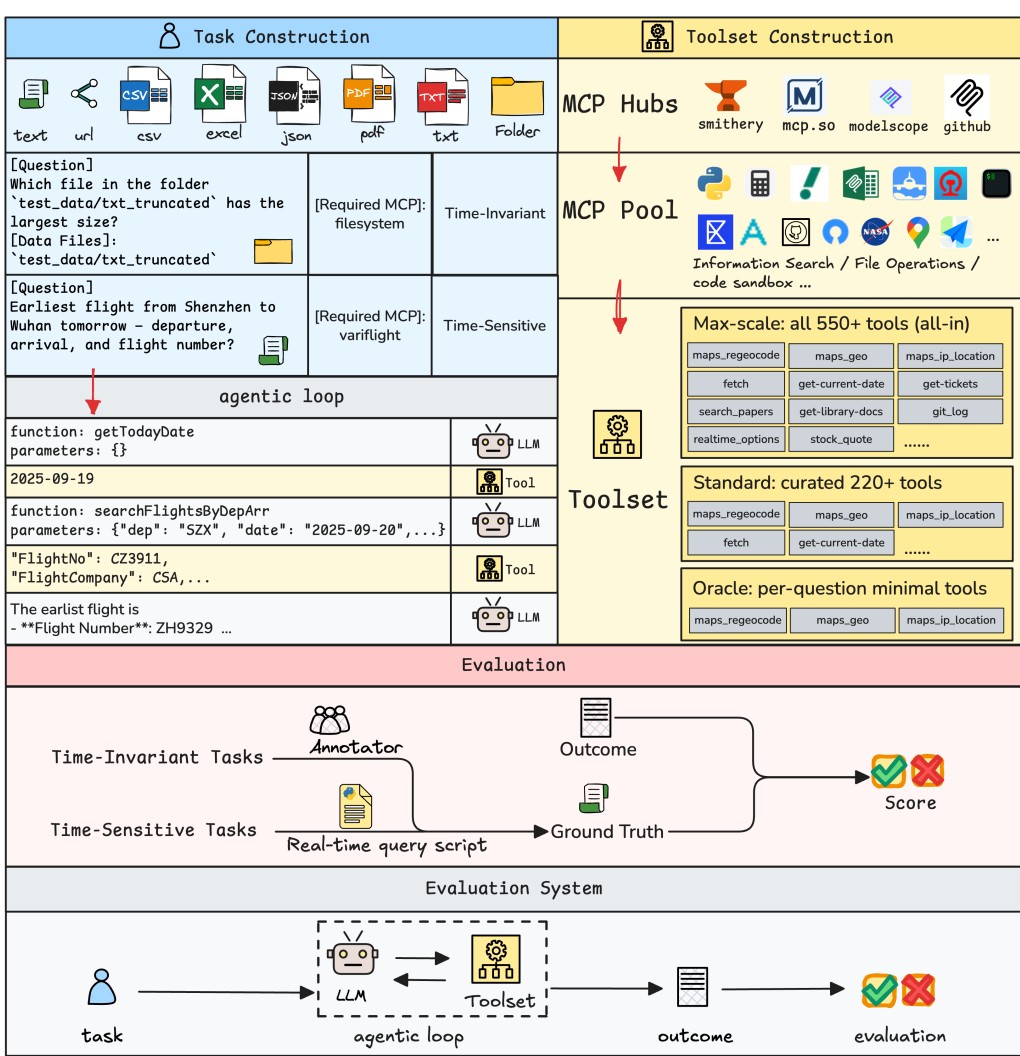

Figure 1: The Overview of MCPVerse. MCPVerse evaluation system. Tasks are curated with metadata and associated data files of various formats. Tools are collected from MCP hubs into an MCP pool and mounted as a toolset under three modes: Oracle, Standard, and Max-Scale. An LLM agent executes an agentic loop—planning and invoking tools via function calls—and produces an outcome. For time-invariant tasks the outcome is compared with human annotations; for time-sensitive tasks real-time scripts retrieve ground truth; an automated scorer then determines correctness.

- Our empirical evaluation of leading state-of-the-art models reveals their performance boundaries and practical limitations when faced with large-scale toolsets. Furthermore, our results demonstrate that a expansive action space provides a tangible benefit for agentic model, rather than merely acting as a hindrance.

## 2 RELATED WORK

### 2.1 AGENTIC TOOL USE

There is a growing interest in equipping LLMs with external tools to overcome their inherent limitations in real-world applications. Fine-tuned models such as Toolformer (Schick et al., 2023), ToolLLaMA (Qin et al., 2024), and Gorilla (Patil et al., 2024) have demonstrated the ability to accurately use different tools like calculators and search engines. In parallel, strategies such as

ToolkenGPT (Hao et al., 2023), METATOOL (Wang et al., 2024b), and IPR (Xiong et al., 2024) have been introduced to enhance the compatibility and effective use of various tools. Furthermore, agentic frameworks like WebMap (Spiegel & Horák, 2024), ReAct (Yao et al., 2023), Voyager (Wang et al., 2024a), Agent Reasoning (Wu et al., 2025), Middleware (Gu et al., 2024), and ViperGPT (Surís et al., 2023) enable models to perform complex, multi-step tasks. These applications range from online shopping and software repair to handling multimodal inputs and learning skills in interactive environments like Minecraft.

## 2.2 TOOL-USE BENCHMARKS

As LLMs are increasingly equipped with external tools to tackle complex real-world applications, the need for robust evaluation of their true capabilities has become critically important. Existing LLM benchmarks have attempted to evaluate model performance in tool-use scenarios, but as agent architectures, MCP, and diverse tool integrations have evolved, a growing gap has emerged between legacy evaluations and LLMs' tool-use capabilities in real-world applications.

Numerous benchmarks have been developed to evaluate the tool-use capabilities of LLMs, each contributing different methodologies. Some efforts, such as ToolAlpaca (Tang et al., 2023) and NexusRaven (Srinivasan et al., 2023), have focused on automatic data generation to create large-scale evaluation sets. Another line of work has concentrated on creating diverse execution environments. For instance, Toolbench (Qin et al., 2024) provides a collection of non-executable APIs, while a more common strategy involves building sandboxed environments to simulate real-world interactions. BFCL-v3 (Patil et al., 2024), for example, implements codebases to simulate services like Twitter and mathematical calculators, and API-Bank (Li et al., 2023b) simulates common tasks such as sending emails or querying stock prices with mock data. Focusing on mobile-centric scenarios, ToolSandbox (Lu et al., 2025) simulates tools for managing device states like Wi-Fi, cellular service, and location service, while HammerBench (Wang et al., 2025) replicates API functionalities based on the features of 60 commercial applications. Other benchmarks have concentrated on specific domains, with ComplexFuncBench (Zhong et al., 2025) targeting travel-related tasks, $\tau$-bench (Yao et al., 2024) focusing on retail and airline scenarios, and ACEBench (Chen et al., 2025) covering four scenarios including food delivery and financial services. Fundamentally, these benchmarks assess model capabilities using simulated, simplified tools; their interaction logic and return values are based on fixed rules or templates, lacking the dynamism and uncertainty of real-world environments, and the complexity of the data they handle does not match that of practical applications.

While previous work has attempted to connect LLMs with real-world APIs, such as in AutoFeedback (Liu et al., 2024; Song et al., 2023), these efforts often faced challenges due to complex and non-standardized interfaces. The Model Context Protocol (MCP) (Anthropic, 2024) was introduced to address this issue. MCP is a standardized protocol for LLM-tool interaction that enables dynamic discovery and orchestration of tools based on task requirements. As the protocol has matured, it has seen widespread adoption across the industry(OpenAI, 2025; cur, 2025; Anthropic, 2025). Consequently, recent work has begun to establish benchmarks on top of MCP. For instance, MCP-Zero (Fei et al., 2025) proposed a retrieval strategy for MCPs and collected an associated toolset, MCP-Tools. Meanwhile, MCPBench (Luo et al., 2025) focused on comparing the performance of different MCPs for web-search tasks, whereas MCP-RADAR (Gao et al., 2025) focused on mathematics and coding tasks and included a subset of general-purpose MCP scenarios.

## 3 MCPVERSE DESIGN

### 3.1 MCP COLLECTION

We collect a diverse set of MCPs from platforms such as Smithery, mcp.so, and ModelScope. These MCPs offer a wide range of functionalities, including local file operations, real-time flight booking, SQL querying and so on. As a Results, we carefully selected MCPs to be included in our benchmark based on the following criteria.

First, to ensure the long-term usability of the benchmark, we prioritize stable MCP services, such as those maintained by well-established organizations or widely adopted by the community. Second, we aim to minimize reliance on services requiring API keys. Excessive API key dependencies

can introduce significant friction for researchers attempting to reproduce or extend our benchmark. Nonetheless, we retain a small number of highly valuable MCPs that do require API keys, in order to preserve the benchmark's practical relevance. Third, we consider the evaluability of each MCP. While some MCP (e.g. Gmail, Notion) are valuable in real-world applications, they are difficult to evaluate automatically. Such MCPs are excluded from task construction but may still be included in the full MCP pool as distractors.

In total, the selected 65 MCPs expose more than 552 individual tools. This number may fluctuate, as MCP providers have the discretion to add or deprecate tools over time.

## 3.2 TASK CURATION

The MCP servers form a large and diverse action space, enabling the execution of a wide variety of task types. Based on this action space, we curate a benchmark consisting of 250 tasks, categorized into two main types: **Information Retrieval** and **System Operation**, and further classified by their complexity, which is divided into three levels: **L1**, **L2**, and **L3**.

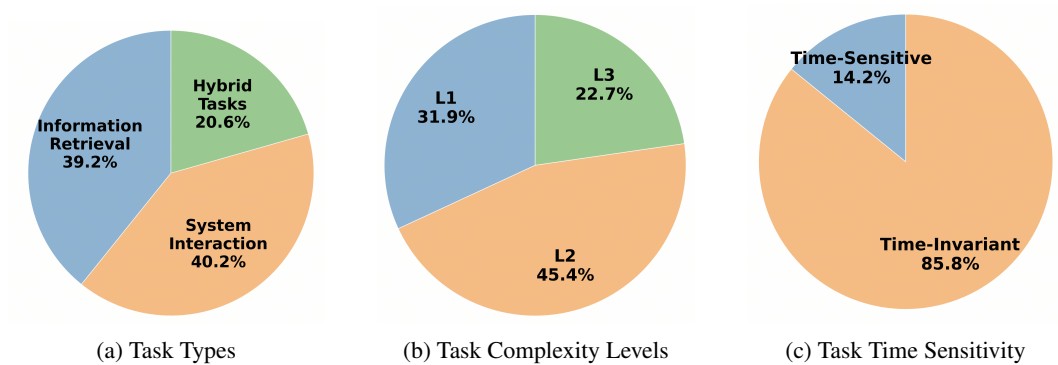

(a) Task Types      (b) Task Complexity Levels      (c) Task Time Sensitivity

Figure 2: Task distributions by (a) task type, (b) task complexity level, and (c) task time sensitivity.

**Task Complexity Levels:** **L1** tasks are solvable by a single tool within 1 or 2 steps. **L2** tasks require about 5 steps and may involve a single tool or multiple tools working together. **L3** tasks are relatively complex and demand multi-tool collaboration or in-depth application of a specific tool, typically requiring more than 5 steps to complete.

**Task Types:** We categorize the tasks into two main types: **Information Retrieval** and **System Operation**. Information Retrieval tasks involve retrieving or querying specific types of information. These include subtypes such as: *Geographical Information* (location-based data), *Financial Information* (market trends, stock prices), *Academic Research* (retrieving academic papers or citations), and *Hot News* (real-time news updates). System Operation tasks include executing system commands (for example, cmd/shell), performing database operations, and handling files. Supported file types include *Text Files* (.txt), *PDF Files* (.pdf), *Presentation Files* (.ppt, .pptx), *Document Files* (.docx), and *Spreadsheet Files* (.xlsx).

**Curation Protocol:** During the task construction phase, annotators (all undergraduate-level or above) select relevant MCPs from the MCP Hub according to the standards outlined earlier, then design tasks based on these MCPs. Annotators thoroughly review each tool definition within the MCPs and ensure the following: (1) A valid solution path exists within the selected set of MCPs. (2) The answers are objective and unambiguous to facilitate LLM-based scoring. (3) Tasks reflect real-world scenarios as closely as possible. (4) Tasks cannot be solved by the model without invoking external tools. Annotators use MCP client utilities (e.g., Cursor) to assist with question generation and answer retrieval. Each task is labeled with the following attributes: question, required MCPs, required tools, time sensitivity, complexity, task type, and ground truth. After initial construction, every task undergoes a comprehensive review. For time-sensitive tasks, custom scripts are employed to retrieve real-time answers for validation.

## 3.3 EVALUATION MODES

Unlike many previous tool-use benchmarks that often incorporate a retrieval module or mount a target tool alongside a small set of unrelated tools (Fei et al., 2025; Li et al., 2023b; Qin et al., 2024), our objective is to significantly expand the toolset available to the LLM. This approach is motivated by two key reasons: (1) Recent advancements in large language models have significantly increased the context window, with models now capable of processing up to 1 million tokens. This alleviates the previous limitation where the toolset size was constrained by context length. (2) We aim to observe how LLMs perform in a large tool space, exploring new solution paths and exhibiting emergent behaviors or novel problem-solving strategies. A retriever trained on existing patterns might in- advertently hinder this crucial exploratory potential.

Based on this, we designed three distinct evaluation modes:

- **Oracle Mode:** In this mode, we load only the minimal set of tools required to solve a given task. This minimal set is defined and annotated by the annotators, who, during the task construction process, verify the feasibility of using these tools to solve the task. As a result, the context length varies depending on the specific task and the tools required.

- **Standard Mode:** This mode is designed for a 64k-token context length, a standard supported by most state-of-the-art LLMs. The toolset in this mode is the union of the minimal tool sets required for each task in Oracle Mode. It consists of 32 MCPs (totaling 220+ tools), whose combined definitions occupy approximately 44k tokens, leaving about 20k tokens for the user prompt and the model's response.

- **Max-Scale Mode:** This mode expands on the Standard Mode by loading all 65 MCPs with 550+ tools simultaneously. The combined context length required is approximately 140k tokens. In our evaluation, only models with extended context capabilities, such as Claude-4-Sonnet and Gemini-2.5-Pro, were able to complete the evaluation successfully, as they can handle the large number of tools and the extensive context length.

## 3.4 EVALUATION PIPELINE

We adopt a simple evaluation pipeline(see Appendix B) with the goal of assessing the LLM's fundamental capabilities, rather than evaluating the effectiveness of complex agentic frameworks like ReAct (Yao et al., 2023) or RolePlaying (Hu et al., 2025). These advanced methods often depend on sophisticated prompt engineering, which can obscure the model's intrinsic abilities. In contrast, our direct approach reduces the impact of such artifacts, yielding a more faithful measure of the model's core agentic capability.

## 3.5 TASK FORMULATION

**Input and Output** Consider a collection of MCP servers $S = \{s_1, s_2, \ldots, s_k\}$, each server $s_i$ providing a toolset $T_i = \{t_{i,1}, t_{i,2}, \ldots, t_{i,N}\}$ accessible via the MCP protocol. Given a task $Q$ and a set of candidate data files $D$, the task in MCPVERSE is to enable an agent $\mathcal{G}$ to address $Q$ by invoking suitable tools from the available pool $T$, which yields an outcome $\mathcal{O}$, i.e., $\mathcal{O} = \mathcal{G}(Q, D \mid T)$, where $\mathcal{O}$ may represent either a final answer or modifications to the associated file states.

**Evaluation Metric** Given MCPVerse's expansive action space, multiple valid action trajectories may exist to accomplish a task. While we cannot directly evaluate every possible trajectory, the final outcome (or end state) must align with the ground-truth specification. Formally, for a task $\tau$ with ground-truth outcome $O$, let $\hat{O} = \mathcal{G}(Q, D \mid T)$ denote the outcome produced by an agent $\mathcal{G}$. We define an evaluation function

$$E(\hat{O}, O) \in \{0, 1\},$$

which returns 1 if the predicted outcome $\hat{O}$ is consistent with the ground truth $O$, and 0 otherwise.

In practice, $E(\cdot)$ is instantiated differently depending on the task type: (i) for tasks with textual ground truth, we employ an LLM and prompt to assess semantic equivalence; (ii) for tasks involving file modifications or state changes, we use a dedicated evaluation script. More details can be found in the Appendix D If $E(\cdot)$ returns 1, the task is considered successfully completed. Aggregating success rates across all tasks yields a quantitative measure of the model's agentic capability.

Table 2: Success rates (SR) of all models in Oracle, Standard, and Max-Scale modes, with break-downs by task levels (L1–L3). SR is the average of L1, L2, and L3. Context Window denotes the maximum input length (in tokens), and Max Tool Count the maximum number of tools an API allows. In the Tool Count column, a dash ($-$) means either no tool cap or a limit larger than the full Max-Scale toolset. In the Max-Scale results, a dash indicates the model could not be evaluated due to context constraints. Scores with $*$ were obtained via prompt-based function calling.

| Model | Context Window | Max Tool Count | Oracle Mode | | | | Standard Mode | | | | Max-Scale Mode | | | |
|---|---|---|---|---|---|---|---|---|---|---|---|---|---|---|
| | | | SR | L1 | L2 | L3 | SR | L1 | L2 | L3 | SR | L1 | L2 | L3 |
| DeepSeek-V3-0324 | 64k | 128 | 46.8 | 62.7 | 45.1 | 32.7 | 32.2 | 47.0 | 27.7 | 22.0 | $-$ | $-$ | $-$ | $-$ |
| DeepSeek-V3.1-Terminus | 128k | 128 | 56.6 | 64.4 | 57.3 | 48.3 | 52.1 | 62.0 | 49.4 | 45.0 | $-$ | $-$ | $-$ | $-$ |
| DeepSeek-R1-0528 | 64k | 128 | 56.4 | 70.5 | 56.4 | 42.4 | 49.9 | 65.1 | 47.4 | 37.3 | $-$ | $-$ | $-$ | $-$ |
| Claude-4-Sonnet | 200k | $-$ | 62.3 | 71.6 | 62.7 | 52.5 | **62.4** | **75.9** | 60.4 | **50.9** | **44.2** | **45.8** | **40.5** | **46.2** |
| Qwen3-235B-A22B | 128k | $-$ | 42.1 | 61.6 | 34.8 | 30.0 | 37.7 | 52.3 | 35.9 | 25.0 | $-$ | $-$ | $-$ | $-$ |
| Qwen3-235B-2507 | 1M | $-$ | 44.8 | 62.5 | 44.8 | 27.1 | 53.2 | 63.9 | 51.8 | 43.9 | 31.6 | 44.3 | 23.7 | 26.7 |
| Qwen3-30B-A3B | 128k | $-$ | 27.7 | 46.5 | 18.1 | 18.3 | 18.9 | 27.9 | 12.9 | 15.8 | $-$ | $-$ | $-$ | $-$ |
| Gemini-2.5-Pro | 1M | 512 | 48.7 | 66.3 | 42.6 | 37.3 | 45.3 | 62.4 | 41.9 | 31.6 | 31.4$^*$ | 37.7$^*$ | 35.7$^*$ | 20.8$^*$ |
| GPT-4o-20241120 | 128k | 128 | 42.1 | 59.1 | 40.2 | 27.1 | 31.4$^*$ | 37.7$^*$ | 35.7$^*$ | 20.8$^*$ | $-$ | $-$ | $-$ | $-$ |
| GPT-5 | 400k | $-$ | **68.1** | **80.5** | **67.8** | **55.9** | 23.4$^*$ | 30.6$^*$ | 19.7$^*$ | 20.0$^*$ | $-$ | $-$ | $-$ | $-$ |
| Kimi-K2-0711 | 128k | 128 | 59.4 | 70.9 | 59.8 | 47.5 | 16.3$^*$ | 24.4$^*$ | 24.4$^*$ | 0.0$^*$ | $-$ | $-$ | $-$ | $-$ |
| GLM-4.5 | 128k | $-$ | 55.0 | 70.9 | 58.5 | 35.6 | 59.1 | 67.4 | **60.7** | 49.2 | $-$ | $-$ | $-$ | $-$ |

# 4 EXPERIMENT

## 4.1 SETUPS

Our evaluation system, built upon the CAMEL framework (Li et al., 2023a) (Figure 1), conducts evaluations of leading LLMs, including DeepSeek-V3, DeepSeek-V3.1-Terminus, DeepSeek-R1-0528, Claude-4-Sonnet, Qwen3-235B-A22B, Qwen3-235B-2507, Qwen3-30B-A3B, Gemini-2.5-Pro, GPT-4o-20241120 and GPT-5, Kimi-K2-0711 and GLM-4.5 across three distinct modes, as described in subsection 3.3. Due to context length limitations, the Max-Scale mode, which requires approximately 147k tokens, can only be evaluated with models such as Claude-4-Sonnet and Gemini-2.5-Pro. Additionally, for scenarios where the toolset exceeds the API's tool number limits but does not exceed the context length—such as GPT-5, which caps the number of tools at 128—we adopt an approach similar to Patil et al. (2025), referred to as **prompt-based function calling**, embedding the complete tool definitions within a custom system prompt (provided in Appendix E), thus bypassing the API's fixed tool number limitations. More setup details see Appendix C.

## 4.2 RESULTS ANALYSIS

This section presents a detailed analysis of the experimental results. Owing to tool-mounting constraints, the Standard mode score for GPT-4o-20241120 and GPT-5 and the Max-Scale mode score for Gemini-2.5-Pro are both obtained via prompt-based function calling.

### 4.2.1 OVERALL MODEL PERFORMANCE

(1) Winners by mode. From Table 2, the best Oracle SR is achieved by GPT-5 (68.1), whereas Standard and Max-Scale are led by Claude-4-Sonnet with 62.4 and 44.2, respectively. GLM-4.5 achieve a very close performance to Claude-4-Sonnet.

(2) MCPVerse is Challenging. On the one hand, as shown in Table 2, even the strongest models achieve modest scores—under Max-scale mode the top model (Claude-4-Sonnet) reaches only 44.2 average accuracy, and on the most difficult L3 subset the best score is just 46.2. On the other hand, many models cannot be run end-to-end under normal settings: Max-Scale demands $\sim$147k tokens and is therefore feasible only for Claude-4-Sonnet and Gemini-2.5-Pro; moreover, several APIs cap the number of tools (e.g., 128 for GPT-4o-20241120/GPT-5 and 512 for Gemini), preventing the full Standard/Max-Scale toolsets from being mounted natively.

### 4.2.2 THE IMPACT OF AN EXPANDED ACTION SPACE

As shown in Fig. 3, models exhibit heterogeneous trends when moving from Oracle to Standard: three improve, whereas the remainder decline, with drops ranging from mild to severe. (1) Improvements trace recent agentic advances. The three gainers—Claude-4-Sonnet, Qwen3-235B-2507, and GLM-4.5—are recent (4-month) releases positioned for agentic use; their higher Standard scores indicate stronger exploration and exploitation of a large-scale tool space, showing that MCPVerse sensitively captures progress in agentic capability. (2) Severe declines align with API tool caps. Models with large drops (GPT-4o-20241120/GPT-5/Kimi-K2-0711)—highlighted by dashed traces—share a common constraint: API tool-number limits prevent native function calling in Standard mode,

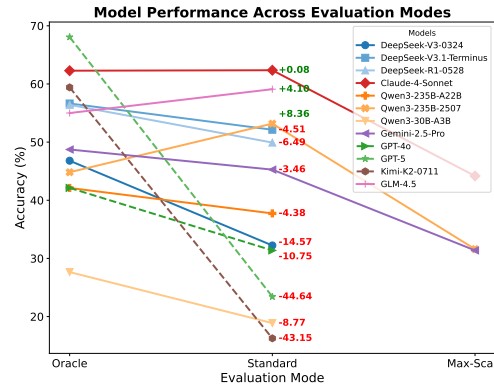

Figure 3: Model performance accuracy across different evaluation modes.

forcing the use of prompt-based function calling. This alternative mechanism likely accounts for their performance degradation and will be discussed in detail in Section 4.2.4. (3) Mild declines reflect context burden. For the remaining models not in (1) or (2), performance degrades as the toolset grows, consistent with heavier schema context and harder tool selection in the larger action space. (4) Larger action spaces remain challenging. For those evaluable at Max-Scale, performance further decreases from Standard to Max-Scale, indicating that even top systems still struggle as the tool space expands.

### 4.2.3 EMERGENT SOLUTION PATHS AND "HACKING" BEHAVIORS

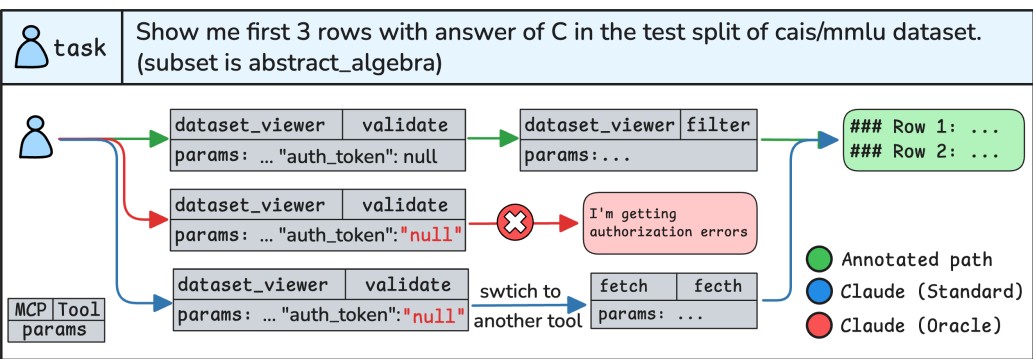

Figure 4: Case study: Claude-4-Sonnet discovers a new solution path. Blocked by a strict authentication format in the limited Oracle mode, the model succeeds in Standard mode by pivoting to an alternative tool textttfetch from the expanded toolset.

Performance gains from Oracle to Standard for several models point to emergent solution paths in larger action spaces: with more tools available and sufficient context budget, models are better able to explore, compose, and exploit alternative tool chains rather than stall on a blocked route. Figure 4 presents a case study in which Claude-4-Sonnet turns an expanded toolset from a potential liability into a clear advantage. The figure depicts three distinct pathways for the same user task. The annotated path (green) is the canonical route, using `validate` and `filter` from the `dataviewer` MCP server. In the constrained Oracle setting (red), Claude-4-Sonnet initially follows this route but fails due to a type mismatch on `auth_token`—it supplies a string instead of the required boolean—triggering an authorization error and causing the attempt to be abandoned. In the Standard setting (blue), after encountering the same initial failure, the model leverages the larger toolset to pivot to `fetch` and completes the task. This switch illustrates the 'hacking' behavior we observe: the model reaches the goal by switching to another solution path and bypassing the interface that had blocked the annotated path.

### 4.2.4 Prompt-Based Function Calling vs. Native Function Calling

To bypass native tool number limitations, we employed a prompt-based function calling method that integrates tool schemas directly into the system prompt. A comparison (Table 3) of this prompt-based method (Prompt) with native function calling (FC) reveals two key findings. First, Claude-4-Sonnet exhibited a significant performance degradation in both Oracle and Standard modes. We attribute this to a substantial mismatch between the function-calling templates in our prompt and those from the model's original training. This discrepancy caused a high rate of hallucination, exceeding 70%, where the

Table 3: Performance comparison between native function calling (FC) and a prompt-based method (Prompt)

| Model | Oracle Mode | | Standard Mode | |
|---|---|---|---|---|
| | FC | Prompt | FC | Prompt |
| Claude-4-Sonnet | 62.28 | 35.55 ↓ | 62.36 | 15.10 ↓ |
| DeepSeek-V3-0324 | 46.80 | 45.79 ↓ | 32.23 | 24.92 ↓ |
| DeepSeek-V3.1-Terminus | 56.64 | 55.05 ↓ | 52.13 | 41.43 ↓ |
| Qwen3-235B-2507 | 44.82 | 49.66 ↑ | 53.18 | 37.99 ↓ |
| Qwen3-235B-A22B | 42.12 | 45.49 ↓ | 37.74 | 26.94 ↓ |
| Qwen3-30B-A3B | 27.65 | 28.86 ↑ | 18.88 | 20.67 ↑ |
| GPT-4o-20241120 | 42.13 | 42.43 ↑ | – | – |
| GPT-5 | 68.06 | 68.28 ↑ | – | – |
| Kimi-K2-0711 | 59.41 | 60.57 ↑ | – | – |
| GLM-4.5 | 55.00 | 52.68 ↓ | – | – |

model frequently fabricated tool responses despite explicit instructions to the contrary. Second, for all other models, the performance difference between the Prompt and FC methods was marginal in the constrained Oracle mode. This suggests that models can effectively execute function calls via system prompts when the number of tools is small. In the more complex Standard mode, however, these same models experienced a sharp performance decline.

### 4.3 Evaluation under Retrieve Mode

In the MCPVerse evaluation system, although the tool space is already large, we initially excluded retrieval because we believed it could restrict the LLM's exploration of the full tool space. In this section, we add experiments under the retrieve mode. Following the semantic similarity-based retrieval methods of Patil et al. (2024) and Moon et al. (2024), we use OpenAI's `text-embedding-3-large` model to embed the descriptions of all MCP servers and store them in a database. During evaluation, each tested model first generates the descriptions of the MCP servers it needs. These descriptions are also embedded with `text-embedding-3-large`, and the top-5 most relevant servers are selected using cosine similarity. These retrieved servers are then mounted to the model for MCPVerse evaluation. The key difference between retrieve mode and oracle mode lies in the toolset that is mounted. In oracle mode, the annotated ground-truth servers are provided, whereas in retrieve mode, more servers may be mounted but the selection is less precise. Under this setting, as shown in Table 4, model performance in retrieve mode is consistently and significantly lower than in oracle mode and standard mode.

Table 4: Comparison of model success rates (SR) across Oracle, Standard, and Retrieve modes.

| Model | Oracle (SR) | Standard (SR) | Retrieve (SR) |
|---|---|---|---|
| DeepSeek-V3.1-Terminus | 56.6 | 52.1 | 39.8 |
| Qwen3-235B-2507 | 44.8 | 53.2 | 39.4 |
| Claude-4-Sonnet | 44.8 | 53.2 | 40.3 |
| GPT-5 | 68.1 | 23.4* | 45.0 |

## 5 Conclusion

In this work, we introduce MCPVerse, a novel benchmark designed to address two critical gaps in prior work: a lack of realism and insufficient scale. Our findings demonstrate that a large-scale action space is surprisingly advantageous for agentic models, as it empowers them to explore a comprehensive set of solution paths to find a working solution. Future work can extend the dataset's scale and scope, as well as investigate model performance using more complex evaluation pipelines.

ACKNOWLEDGMENTS

We thank SenseTime for providing computing resources. We also thank MCP providers and CAMEL developers for their support.

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

## A  APPENDIX

## B  EVALUATION PIPELINE

```python
toolset = []
for mcp in mcp_list:
    mc = connect_to(mcp)
    toolset.append(mc.get_tools())

task = get_user_input()
history = [task]

while True:
    resp = LLM(history, toolset)
    history.append(resp)
    if not resp.tool_calls:
        return

    for tool_call in resp.tool_calls:
        result = call_tool(tool_call)
        history.append(result)
```

## C    HYPERPARAMETERS

This section details the specific hyperparameter configurations used throughout the experiments. The hyperparameter configurations for each model, as detailed in Table 5, were selected based on a consistent methodology. For models where the developers provide official guidance on optimal generation parameters, such as the Qwen3 series and Kimi-K2-0711, we adopted these recommended settings. In the absence of specific recommendations, we applied a uniform temperature of $0.1$. This conservative value was chosen to minimize output randomness and enhance the determinism of the results, a critical factor for ensuring the stability and reproducibility of our experiments.

| Model | Temperature | Top $p$ |
|---|---|---|
| DeepSeek-V3-0324 | 0.1 | — |
| DeepSeek-R1-0528 | 0.1 | — |
| Claude-4-Sonnet | 0.1 | — |
| Qwen3-235B-A22B | 0.7 | 0.8 |
| GPT-4o-20241120 | 0.1 | — |
| Qwen3-30B-A3B | 0.7 | 0.8 |
| Gemini-2.5-Pro | 0.1 | — |
| Kimi-K2-0711 | 0.6 | — |

Table 5: Generation parameter settings for each evaluated model. An em-dash "—" indicates that the parameter was left at the model's default value.

## D    EVALUATION METRIC

Table 6 presents the prompt used to score tasks with textual ground truth.

---

**Prompt used for scoring**

You are a grading assistant. Your task is to evaluate whether a model output and a reference answer are semantically consistent. Please note: the expressions do not need to be exactly the same—so long as the meanings are equivalent, they should be considered consistent.

If they are consistent in meaning, output the score in the following JSON format and explain your reasoning first:

```
{
    "score": 1
}
```

If they are not consistent in meaning, output:

```
{
    "score": 0
}
```

The following is the question, model output and reference answer:

<question>
{question}

<reference>
{answer}

<model output>
{pred}

now give your judgement.

---

Table 6: Prompt templates for the scoring. The placeholders {question}, {answer}, and {pred} are replaced at runtime with the task's question, its ground-truth answer, and the model's predicted answer, respectively.

The following code snippet presents the evaluation script for task Q101, which verifies that a given directory exists, contains `alpha.txt` and `beta.txt`, and that their contents are "Hello Alpha" and "Hello Beta," returning 1 only if all checks pass.

```python
# eval_scripts/Q101.py

import os

def run_test(pred, dir_path) -> bool:
    """
    Verifies that the directory `dir_path` exists and contains:
    - alpha.txt with content 'Hello Alpha'
    - beta.txt with content 'Hello Beta'
    Returns True if all checks pass, False otherwise.
    """
    alpha_path = os.path.join(dir_path, "alpha.txt")
    beta_path = os.path.join(dir_path, "beta.txt")

    # Check if directory exists
    if not os.path.isdir(dir_path):
        return False

    # Check if both files exist
    if not os.path.isfile(alpha_path) or not
    ↪  os.path.isfile(beta_path):
        return False

    # Read and verify content of alpha.txt
    with open(alpha_path, "r", encoding="utf-8") as f:
        alpha_content = f.read().strip()
    # Read and verify content of beta.txt
    with open(beta_path, "r", encoding="utf-8") as f:
        beta_content = f.read().strip()

    print(alpha_content, beta_content)
    return alpha_content == "Hello Alpha" and beta_content ==
    ↪  "Hello Beta"
```

# E    PROMPT-BASED FUNCTION CALLING

This section details the prompts used for guiding evaluated models in performing prompt-based function callings (Table 7).

---

System prompt template for prompt-based function-calling approach

---

You are an expert in composing functions. You are given a question and a set of possible functions. Based on the question, you will need to make one or more function/tool calls to achieve the purpose. Please note that the task may be very complicated. Do not attempt to solve the task by single step. You should only return the function calls in your response until you know the exact answer of the problem. For each time you call a function, the user will return the corresponding execution result in the format `<tool_response>{{result}}</tool_response>`. You can use the result to make further function calls.

<tips>
1. Put all calls in a single list of JSON objects, with the format:

```
[
    {
        "name": "function_name_1",
        "parameters": {
            "param_1A": value1A,
            "param_1B": value1B,
            ...
        }
    },
    {
        "name": "function_name_2",
        ...
        }
    },
    ...
]
```

Never output multiple separate lists or include any text before or after the list.
2. Do not include any other text alongside the list or use other formats. Do not mock any responses of the functions—just return the function call.
3. Your task may span multiple rounds of function calls. If any function depends on results of previous calls, you must first obtain the prerequisite result before invoking the dependent function.
4. If one way fails to provide an answer, try other ways or methods. The answer does exist.
5. When looking for specific numerical values (e.g. dollar amounts), prioritize reliable sources and avoid relying only on search snippets.
6. You can write Python code to solve the task if needed.
7. If you generate any files, please place them in the `outputs` folder.
8. Some tools need internet access. If you encounter connection issues while using them, try retrying the request.
9. If the problem is fully solved and no further function calls are needed, return a concise text summary of the final answer instead of a function list. Otherwise, always return a function list.
</tips>

You are provided with function signatures within <tools></tools> XML tags:
<tools>**{tools_info_list}** </tools>

---

Table 7: System-prompt template for our prompt-based function-calling scheme. At runtime the placeholder {tools_info_list} is populated with the descriptions and parameter schemas of the tools required, which vary by evaluation mode.

## F  ANALYSIS OF JUDGE MODEL SELECTION

This section analyzes the influence of different judge models within the LLM-as-a-judge framework. Our primary evaluation was conducted using the open-source model QwQ-32B as the judge. To analyze the impact of this specific model choice, we conducted a comparative analysis using the another model, GPT-4o, as an alternative judge. The results of this comparison are presented in Table 8.

While minor discrepancies were observed in the detailed scores assigned by the two models, the overall performance rankings and trends of the evaluated results remained consistent. This consistency validates the reliability of using open-source models for evaluation and underscores the quality of our dataset.

| Model | GPT-4o Scoring | | | | | QwQ-32B Scoring | | | | |
|---|---|---|---|---|---|---|---|---|---|---|
| | Rank | SR. | L1 | L2 | L3 | Rank | SR. | L1 | L2 | L3 |
| Claude−4−Sonnet | **1** | **61.26** | **74.59** | 59.20 | **50.00** | **1** | **62.36** | **75.86** | 60.36 | **50.85** |
| GLM−4.5 | 2 | 58.28 | 66.39 | **60.00** | 48.44 | 2 | 59.10 | 67.44 | **60.71** | 49.15 |
| Qwen3−235B−2507 | 3 | 51.61 | 61.48 | 49.60 | 43.75 | 3 | 53.18 | 63.86 | 51.82 | 43.86 |
| DeepSeek−V3.1−Terminus | 4 | 51.07 | 60.66 | 48.80 | 43.75 | 4 | 52.13 | 62.03 | 49.35 | 45.00 |
| DeepSeek−R1−0528 | 5 | 49.02 | 63.93 | 47.20 | 35.94 | 5 | 49.92 | 65.12 | 47.37 | 37.29 |
| Gemini−2.5−Pro | 6 | 44.49 | 60.66 | 40.00 | 32.81 | 6 | 45.27 | 62.35 | 41.88 | 31.58 |
| Qwen3−235B−A22B | 7 | 37.29 | 52.46 | 34.40 | 25.00 | 7 | 37.74 | 52.33 | 35.90 | 25.00 |
| DeepSeek−V3−0324 | 8 | 31.93 | 46.72 | 27.20 | 21.88 | 8 | 32.23 | 46.99 | 27.68 | 22.03 |
| GPT−4o−20241120 | 9 | 30.27* | 36.89* | 33.60* | 20.31* | 9 | 31.38* | 37.66* | 35.71* | 20.75* |
| GPT−5 | 10 | 23.52* | 28.69* | 20.00* | 21.88* | 10 | 23.42* | 30.59* | 19.66* | 20.00* |
| Qwen3−30B−A3B | 11 | 19.29 | 27.87 | 12.80 | 17.19 | 11 | 18.88 | 27.91 | 12.93 | 15.79 |
| Kimi−K2−0711 | 12 | 17.02* | 27.05* | 24.00* | 0.00* | 12 | 16.26* | 24.42* | 24.35* | 0.00* |

Table 8: Accuracy results for all evaluated models in Standard mode, scored by GPT-4o and QwQ-32B. Ranks are based on the average scores for each judge model. Scores marked with an asterisk * were obtained using prompt-based function calling rather than native tool use due to context-length limits.

## G  THE USE OF LARGE LANGUAGE MODELS

We used a large language model (GPT-5, OpenAI; accessed September 2025) only for copyediting. The model suggested edits to grammar, wording, and style. It did not generate ideas, write technical content, produce code, design experiments, run analyses, or create figures or tables. All edits were reviewed by the authors, who remain responsible for the final text and claims.

As input to the model, we provided only passages from our own drafts of this paper. We did not share private data or unpublished results. The model did not have access to any dataset used in our study. The use of the model had no effect on the design of experiments, the results, or the conclusions. This disclosure is included to meet venue guidelines.

