# OpenReview forum: "MCPVerse: An Expansive, Real-World Benchmark for Agentic Tool Use"
_ICLR.cc/2026/Conference — Submitted to ICLR 2026_

### Official Review · Reviewer_mGrE · 2025-10-19

**Soundness:** 3
**Presentation:** 3
**Contribution:** 2
**Rating:** 4
**Confidence:** 2

**Summary:**

This paper introduces MCPVerse, a large-scale, real-world benchmark and evaluation system to evaluate the agentic tool use capabilities of LLMs.
The benchmark aggregates large-scale real-world tools and real execution system via MCP and support multiple evaluation modes  (Oracle, Standard, Max-Scale), hybrid outcome-based evaluation, and automated assessment pipelines.
Experiments with the best LLMs highlight practical challenges and expose current limitations, especially in handling large toolsets at scale.

**Strengths:**

- MCPVerse uses real-world tools to evaluate the performance in real environments. Even the best LLMs cannot solve most of the tasks.

- MCPVerse has much more tools than the previous benchmarks and discover the effect of different action spaces.

- Section. 4 is detailed and brings new insights about LLM tool-use. It calls the larger tool number limitation for real-world application of future LLMs.

**Weaknesses:**

- There are a total of 250 tasks, and most tasks (L1 & L2) require fewer than five steps. I am concerned whether most tools have been sufficiently utilized.
- The tasks are manually designed, which ensures naturalness and reliability, but this approach may not be sustainable.
- The changes of remote MCP servers may change the evaluation results.

**Questions:**

See the "weaknesses".

---

### Official Review · Reviewer_6SCH · 2025-10-31

**Soundness:** 3
**Presentation:** 3
**Contribution:** 3
**Rating:** 6
**Confidence:** 4

**Summary:**

MCPVerse is a large-scale, real-world benchmark for evaluating agentic tool use, designed to address two major limitations of existing benchmarks: reliance on synthetic tools (existing benchmarks predominantly use simulated tools, lacking realism) and constrained action space (due to context length limitations, typically only a small number of tools can be mounted). MCPVerse integrates 65 MCPs, 552 tools, and 250 tasks, featuring three evaluation modes with different action space scales and three complexity levels. The evaluation focuses on outcome-based assessment rather than specific tool invocation trajectories. MCPVerse reveals a crucial insight: an expanded action space is advantageous rather than detrimental for SOTA agentic models, as they can leverage the extended toolset to explore diverse solution paths.

**Strengths:**

1. The benchmark integrates 65 MCPs with 552 tools, with tool definitions exceeding 147k tokens. This is a leap from prior work that relied on simulated or mock tools, bringing evaluations much closer to real-world deployment scenarios.

2. The 250 tasks span realistic scenarios across information retrieval and system operations, with a three-level complexity taxonomy. The inclusion of time-sensitive tasks with real-time ground truth validation is particularly noteworthy.

3. The three evaluation modes (Oracle, Standard, Max-Scale) provide a systematic way to probe model capabilities under varying action space scales. This design choice is well-motivated and allows to study how model performance changes with toolset complexity.

4. The hybrid outcome-based evaluation approach is sensible—using LLM-as-a-judge for semantic consistency and automated scripts for verifying state changes. Importantly, the focus on final outcomes rather than prescribed tool-calling trajectories avoids penalizing models for finding alternative valid solutions.

5. The experiment is comprehensive and insightful, covering 12 models across different modes. The experiments reveal a counter-intuitive finding: certain agentic models (Claude-4-Sonnet, Qwen3-235B-2507, GLM-4.5) actually improve when given access to larger toolsets, demonstrating emergent solution paths. The analysis of prompt-based vs. native function calling, retrieval-based approaches, and various model limitations adds significant depth.

6. The paper is well-written with clear figures (especially Figure 1 and Figure 4) and comprehensive tables. The methodology is described in sufficient detail for reproducibility, and the appendices provide thorough documentation of implementation details.

**Weaknesses:**

Refer to questions section.

**Questions:**

**Question 1: Counter-intuitive difficulty ranking in Max-Scale Mode**

In Max-Scale Mode, both Claude-4-Sonnet and Qwen3-235B-2507 exhibit a phenomenon where L2 success rates are lower than L3 success rates. Does this counter-intuitive pattern suggest that the difficulty rating system may be biased or miscalibrated? Could the authors explain the underlying causes of this phenomenon and provide specific case studies demonstrating why certain L3 tasks might be easier to solve than L2 tasks in the Max-Scale setting?

**Question 2: Performance gains from prompt-based function calling**

PROMPT-BASED FUNCTION CALLING shows performance improvements for certain models in both Oracle and Standard modes. Does this suggest that the prompt-based method provides certain advantages in tool selection or reasoning that native function calling lacks? Or does this indicate that the native function calling APIs of these models are not well-optimized? Could the authors explain the causes of these counter-intuitive performance gains and provide specific case studies illustrating when and why prompt-based function calling outperforms native approaches?

**Question 3: Source of the 70% hallucination rate**

In line 449 (Section 4.2.4), the paper states: "This discrepancy caused a high rate of hallucination, exceeding 70%, where the model frequently fabricated tool responses despite explicit instructions to the contrary." How was this 70% hallucination rate calculated? Could the authors provide more details on the methodology used to compute this metric and whether similar hallucination rates were observed in other models?

**Question 4: Exploring retrieval performance with varying top-K**

Regarding EVALUATION UNDER RETRIEVE MODE (Section 4.3), the current top-5 retrieval limit may be the primary cause of the poor performance compared to Standard mode (which uses 32 MCPs with 220+ tools). In fact, the fixed use of large MCP sets, though potentially yielding higher performance, occupies substantial context that is infeasible in practice. Exploring how to improve the performance of dynamically retrieved MCPs may be a more promising direction with greater practical value.

Could the authors show how model performance changes as top-K increases? Specifically, is there a value of K (smaller than 32) where performance approaches that of Standard Mode (which contains 32 MCPs and 220+ tools)? This analysis would help identify whether the poor Retrieve mode performance stems from insufficient recall (top-5 being too small) or from fundamental limitations in the retrieval method itself. Understanding the performance-context tradeoff curve would provide valuable insights for practical deployment scenarios.

---

### Official Review · Reviewer_cTyv · 2025-10-31

**Soundness:** 2
**Presentation:** 2
**Contribution:** 2
**Rating:** 2
**Confidence:** 5

**Summary:**

The paper proposes "LLMs are becoming tool-using agents, but current benchmarks rely on synthetic tools and tiny action spaces. MCPVerse introduces a real-world benchmark with 550+ executable tools, an action space >147k tokens, and outcome-based, real-time evaluation across Oracle, Standard, and Max-Scale modes. Results show most models degrade as tool sets grow, while agentic models (e.g., Claude-4-Sonnet) can exploit larger tool spaces—highlighting both current limitations and MCPVerse’s value." The main contributions of this paper are,

- MCPVerse: A real-world tool-use benchmark with 550+ executable tools and carefully designed tasks, each with ground truth (or scripts for time-sensitive cases) and metadata like required MCPs, difficulty, and task type.

- Evaluator: An end-to-end automated system for multi-step agent–tool interactions, scoring final outcomes with a hybrid metric—LLM-as-judge for text plus scripts to verify state changes.

- Findings: Testing SOTA models reveals clear limits with large toolsets; importantly, bigger action spaces help agentic models rather than just hindering them.

**Strengths:**

The main strengths of this paper can be summarized as,

- The article is written in plain, easy-to-understand language.

- The proposed dataset size is acceptable.

**Weaknesses:**

The  main weaknesses and questions of the paper are listed below,

- The issues in Introduction section:

    - Over-generalization of “artificial tools.” Claiming “many benchmarks rely on artificial tools” ignores widely used real-execution suites (e.g., SWE-bench runs tests on real repos; WebArena/BrowserGym execute live web actions; OpenDevin/AgentBench run OS/terminal tools). The statement needs qualifiers or counter-examples. Meanwhile, the abundance of prior similar work significantly diminishes the paper’s novelty and value.

    - “Stop short of actual execution.” Saying benchmarks that include real APIs usually avoid execution is factually shaky—several do execute API calls or environment steps and verify outcomes. Provide specific citations that truly lack execution, or soften the claim.

    - Evaluation “confined to tool name/parameters.” Many benchmarks score functional outcomes (tests pass, task success on websites, file created, answer matches). Portraying evaluation as name/arg matching is misleading.

    - Unproven causality (“superficial patterns → success”). The text asserts models succeed via pattern matching rather than planning without evidence. Either cite studies demonstrating this failure mode or rephrase as a hypothesis.

    - Conflation of action space with context length. Equating “large API list” with tokens in the prompt is a category error: tool registries can be programmatic/dynamic; agents can retrieve tool specs on demand. Context limits don’t force a tiny tool set.

    - Unsubstantiated quantities. “a few dozen relevant options per query” is asserted without data; some frameworks surface 5–10, others allow hundreds via paging/ID lookups. Provide measured ranges or drop the number.

    - “Artificial tools,” “real-world APIs,” and “action space” aren’t operationalized, making several claims unfalsifiable. Define them before arguing limitations.

- The issues in MCPVERSE DESIGN section:

    - Key-policy conflict. The authors “minimize reliance on API keys” for reproducibility but also “retain a small number of highly valuable MCPs that do require API keys.” This is contradictory and needs a clear policy (e.g., provide test keys, mock servers, or a keyed vs. keyless track).

    - Drift risk. The authors acknowledge providers “add or deprecate tools over time,” yet claim “long-term usability.” Without version pinning/snapshotted specs, the benchmark isn’t reproducible across time.

    - Distractor design. Excluding Gmail/Notion from task construction but keeping them as distractors may unfairly penalize agents that (reasonably) select them; also many such tools require OAuth/keys, so they’re non-actionable distractors.

    - Counting tools. “65 MCPs expose 552 tools” likely includes overlapping/aliased endpoints across providers. If duplicates aren’t deduplicated or typed, the reported action-space size can be misleading.

    - Real-time flight booking. Executing live bookings can incur costs and create ethical/compliance issues. The authors should state that only sandbox/test endpoints are used and how the authors prevent charges or personal-data handling.

    - Minor but impactful wording issues: “As a Results” → “As a result.”; “some MCP (e.g., Gmail, Notion) are …” → “some MCPs … are …”; Ensure consistent capitalization (MCP/tool names) and commas (“e.g., …”).

    - Category mismatch. Text says tasks are in two main types (Information Retrieval & System Operation), but Fig. 2a shows three (adds “Hybrid”) and the label uses “System Interaction.” Define types consistently and say whether “Hybrid” is a third class or a bookkeeping tag.

    - Percentages vs. total count. The authors claim 250 tasks, yet the one-decimal percentages imply non-integer counts (e.g., 40.2% of 250 = 100.5 tasks; 14.2% = 35.5). Report integer counts or percentages that sum from integers.

    - Time-sensitive ground truth ≠ reproducible. The authors “retrieve real-time answers for validation,” which means results change over time; this undermines comparability and “long-term usability.” The authors need timestamped snapshots, cached labels, or versioned replay logs.

    - Complexity definition is circular/confounded. L1/L2/L3 are defined by step counts and number of tools (e.g., “about 5 steps,” “multi-tool”). That makes complexity a property of the chosen solution path/annotation, not the underlying task, and confounds tool routing with difficulty. The authors should measure complexity from execution traces (median steps, tool entropy) rather than prescribe it.

    - “Must require external tools” is unverified. Many IR tasks (e.g., academic facts) can be answered from parametric knowledge. There’s no baseline check showing no-tool models fail these items. Without dynamic/ephemeral instances, this requirement isn’t guaranteed.

    - Annotator protocol leaves quality gaps. “All undergraduate-level or above” annotators, using an IDE assistant (Cursor), but no inter-annotator agreement, adjudication policy, or qualification tests are reported. For financial/news/OS tasks, domain competence matters.

    - Outcome clarity for IR subtypes. Subtypes like “Academic Research (retrieving papers/citations)” can hit paywalls or return non-deterministic results across providers. The page doesn’t state which APIs/endpoints are allowed, how duplicates are handled, or how success is verified (exact DOI match? title?).

    - Security & sandboxing for system operations. The authors allow “executing system commands” and file ops but don’t specify the sandbox/permissions model. Without isolation, evaluations are unsafe or non-portable.

    - “As retrievers may hinder exploration” is asserted without data. Prior work often shows retrieval helps by reducing distraction. Provide ablations (full list vs. retriever) or soften.

    - “44k tokens for definitions, ~20k left for prompt+response (64k window).” This couples task solvability to vendor token limits rather than capability. It also ignores multi-turn overhead. The authors need a fixed, model-agnostic tool-description budget (e.g., 80–120 tokens/tool) or a paging API.

    - Only models with huge windows (Claude-4-Sonnet, Gemini-2.5-Pro) “complete the evaluation.” Results will primarily reflect context capacity, not tool-use skill. That hampers comparisons with mainstream/open-source models.

    - The text cites “up to 1M tokens,” but the benchmark actually uses ~140k. Long-context quality often degrades; without showing retention/attention stability, the premise that “bigger window → better tool use” is speculative. The authors motivate three modes but don’t state ablations (e.g., same tasks under Oracle vs. Standard vs. Max-Scale) to quantify how action-space growth changes success independent of context size.

- Limited analysis in Experiments Section. Unexplained asterisk. The “23.4*” for GPT-5 lacks a footnote/explanation (e.g., context limit, truncation). Claim contradicts Table 4. Text says retrieve-mode is “consistently and significantly lower than oracle and standard,” but for GPT-5 it’s higher than Standard (45.0 vs 23.4*). Reproducibility/fairness. Using OpenAI text-embedding-3-large (both index and query) is closed and vendor-specific; release frozen embeddings or ablate with open models (e.g., bge-large, e5). Retrieval confounded with model self-description. The LLM must first generate tool descriptions; failure to name or phrase them well sinks recall, so results mix retrieval quality with model prior/tool-naming knowledge. Evaluate retrieval with task-derived queries and report recall@k for required servers.

Overall, this paper is unsatisfactory. First, there are many ambiguous statements, and numerous claims are unsupported or unreliable. Second, the dataset construction section is overly brief and lacks sufficient detail for reproducibility. Third, if the appendix is meant to be important, its organization and formatting need careful revision. Finally, including an acknowledgments section at submission risks disclosing the authors’ collaborating institutions (compromising anonymity).

**Questions:**

Please refer to the Weaknesses section. Overall, the paper is unsatisfactory: the motivation shows limited novelty, many critical details are missing, and the experimental design and analysis are sparse. I encourage the authors to thoroughly revise the manuscript.

---

> ### Author Response · Authors · 2025-11-25
>
> **(Part 1/2)**
>
> We sincerely thank you for your time, effort, and thoughtful comments in reviewing our work. In the following, we respond to the main points raised by the reviewer.
>
> **Weakness**
>
> **Introduction section**
> > **“Artificial tools,” “real-world APIs,” and “action space” aren’t operationalized**
>
> - In this paper, “Artificial tools” refer to functions mocked via Python or similar means—for example, in BFCL and API-Bank, tools such as booking flights or logging into Twitter are implemented as mocked functions without performing real actions.
> “Real-world APIs” contrast with these mocked tools; they are actual APIs, and our selected MCPs are chosen specifically to construct such a set. “Action space” mainly refers to the number of available tools.
>
> > **Claiming “many benchmarks rely on artificial tools” ignores widely used real-execution suites**
>
> -  We primarily regard MCPVerse as a function-calling benchmark; accordingly, its main points of comparison are ToolBench, BFCL, and API-Bank (as shown in Table 1), rather than agent benchmarks that focus on a specific domain. Compared with the latter, function-calling benchmarks are not tied to a particular domain and instead focus on the accuracy of general-purpose function calls.
>
> - The details regarding executability and the use of artificial vs. real-world tools in prior benchmarks are already summarized in Table 1. In the Executable Tools column, a total of 0 indicates that the benchmark does not perform actual execution, such as ToolBench. The Real and Simulated columns show the relative quantities of real-world tools versus artificial (mocked) tools. As shown in the table, all pre-MCP benchmarks rely entirely on simulated tools, such as BFCL, API-Bank, ToolSandBox.
>
> > **Conflation of action space with context length.... Context limits don’t force a tiny tool set.**
> -  We thank the reviewer for the clarification. As noted above, our use of “action space” refers specifically to the number of tools available to the model. While agents can indeed employ retrieval-based methods to dynamically load tool specifications, this approach depends heavily on the effectiveness of the retrieval mechanism. In our main experimental setup, tools are directly provided to the model, so a larger tool set naturally occupies more context length. For completeness, Section 4.3 also reports results under the retrieval setting.
>
> > **novelty**
> - We would like to emphasize that our main contribution is MCPVerse, which scales the number of executable real-world tools to a new scale. This larger toolset allows us to observe that certain agentic models actually improve when given access to richer tool collections, revealing emergent solution paths that do not appear at smaller scales.

---

> ### Author Response · Authors · 2025-11-25
>
> **(Part 2/2)**
> **MCPVERSE DESIGN section**
> > **Key-policy conflict. & Drift risk && Time-sensitive ground truth ≠ reproducible**
> - We thank the reviewer for raising this point. We agree that these issues arise naturally when working with real-world tools, but we believe such tools are essential for evaluating models in realistic settings.
>
> - A core design principle of MCPVerse is to assess a model’s ability to solve real-world problems, and therefore we include certain widely adopted MCPs (e.g., flight or routing services) even when they require API keys. In line with this goal, we also introduce time-sensitive tasks and provide carefully designed ground-truth fetching scripts.
>
> > **Counting tools. ...the reported action-space size can be misleading.**
> -  We have explicitly checked for duplicate or aliased endpoints and confirmed that this situation does not occur. The complete deduplicated tool list is provided in the supplementary zip file(`mcpverse/data/mcpverse_benchmark2.xlsx`)
>
> > **Real-time flight booking**
> - The benchmark includes only flight-query tasks, not flight-booking tasks, and no live bookings or payments are ever executed.
>
> > **Minor but impactful wording issues & Category mismatch**
> - Thank for pointing out these wording issues, and we will correct them in the revised version.
>
> > **Percentages vs. total count**
> - We thank the reviewer for pointing this out. The final benchmark does not contain exactly 250 tasks, and we will update the manuscript.
>
> > **“Must require external tools” is unverified.**
> - As described in our Curation Protocol (Section 3.2), we explicitly ensure that all included tasks cannot be reliably solved by the model without invoking external tools.
>
> > **Annotator protocol leaves quality gaps... For financial/news/OS tasks, domain competence matters.**
> - After the initial construction by annotators, all tasks were manually checked by the authors for correctness. The tasks are designed so that they do not require highly specialized domain knowledge. The full details are available in the supplementary zip file, at `mcpverse/data/mcpverse_benchmark2.xlsx`
>
> > **As retrievers may hinder exploration” is asserted without data**
> -  In our experimental setup, both the oracle and standard modes expose all tools necessary to solve each task, whereas the retriever can fail to surface required tools, which can reasonably hinder exploration. Consistent with this expectation, Section 4.3 shows that the retriever mode performs worse than both the oracle and standard modes.
>
> > **The authors need a fixed, model-agnostic tool-description budget (e.g., 80–120 tokens/tool) or a paging API.**
> - The tool sets we provide are model-agnostic: in standard mode the model receives 220+ tools, and in max-scale mode 550+ tools, regardless of vendor-specific context limits.
>
> > **Only models with huge windows (Claude-4-Sonnet, Gemini-2.5-Pro) “complete the evaluation.**
> - All evaluated models are able to complete the benchmark in both the oracle and standard modes, and these scores already reflect their tool-use capabilities.
>
> > **The authors motivate three modes but don’t state ablations (e.g., same tasks under Oracle vs. Standard vs. Max-Scale) to quantify how action-space growth changes success independent of context size.”**
> - **This is exactly what we do**: the Oracle, Standard, and Max-Scale modes all use the same set of tasks. The only difference across modes is the size of the exposed tool set.
>
> > **Limited analysis in Experiments Section.**
> - Scores marked with “*” were obtained using prompt-based function calling; we will add an explicit explanation in the revised manuscript.
> - The higher retrieve-mode score for GPT-5 compared to its standard-mode score is due to the fact that GPT-5 runs the standard mode via prompt-based function calling, which degrades its performance in this setting (as discussed in Section 4.2.4). Apart from this implementation-specific effect for GPT-5, all other results are consistent with our claim that retrieve mode is consistently and significantly lower than both oracle and standard.
> - The technical details of retrieval are not the focus of this work, and therefore Section 4.3 does not expand on retrieval-specific design choices.

---

### Official Review · Reviewer_UkF5 · 2025-11-01

**Soundness:** 2
**Presentation:** 3
**Contribution:** 2
**Rating:** 2
**Confidence:** 4

**Summary:**

The paper introduces a new benchmark for agentic tool use called MCPverse. This benchmark includes 65 MCPs and 552 tools, with a variety of task types, complexities, and time sensitivities. The authors evaluate models from a variety of providers and share the results across three modes (Oracle, Standard, Max-Scale).

**Strengths:**

Relevance: MCPs are gaining popularity and measuring agentic tool use in the context of MCPs makes a lot of sense and is timely.
Unsaturated: The maximum score on the benchmark is still <70%, so the benchmark is far from saturated.
Model evaluation: A diversity of models are evaluated which is great to see. The specific evaluation of large context is also useful.
Judging: Model scores seem relatively stable across judge models (e.g., GPT-4o vs QwQ).

**Weaknesses:**

Sharing tasks: This benchmark would benefit substantially from a more thorough discussion of example tasks so that quality of the evaluation can be better understood. The task types, complexity, and sensitivity are included in the paper, but it is not possible to vet, for example, whether the geographical information retrieval tasks are high-quality unless we get more detail. Moreover, details on the review process to ensure quality are not discussed (beyond the 1 sentence of "After initial construction, every task undergoes a comprehensive review"); we are missing key details on rounds of review, qualifications of reviewers, etc. which are required to verify that the quality is worthwhile. Moreover, full examples of the tasks are never given; a diagram of actual example tasks or a link to a dataset viewer would be much more useful than the pie charts shown.  Finally, open-sourcing/sharing the tasks, even if just a subset (while keeping another subset held-out), would be important for this work to advance the field of benchmarking. This is the biggest current gap in the paper that prevents me from 1) verifying the tasks are actually of high quality and 2) sizing the magnitude of this contribution as substantial. I would be open to reconsidering my review if this detail was provided and the tasks were possible to inspect directly.
Evaluation settings: Much of the evaluation is dependent on the API settings used (e.g., tool cap from the API provider) and this prompt-based function calling workaround; it's hard to tell whether the results are due to limitations of the API endpoint or the scaffold versus the models themselves.
Ease of running: This benchmark seems quite difficult to run and sensitive to scaffold; combined with the lack of open-sourcing of the benchmark, it is not possible to run or reproduce to better understand.
Metrics: The binary evaluation metric for success is a bit confusing -- is there a reason we didn't consider partial credit?
Typos: There are also some small typos, e.g., "Hybird" instead of hybrid on page 2, "As a Results" on page 4.

**Questions:**

What were the settings used for each model? (e.g., GPT-5: what was the reasoning effort?) This could also go in appendix.

The authors mention "After initial construction, every task undergoes a comprehensive review." --> what was this review, and how can we trust the quality?

---

> ### Author Response · Authors · 2025-11-24
>
> We sincerely thank the reviewer for their thoughtful feedback and provide our responses below:
>
> > **This is the biggest current gap in the paper that prevents me from 1) verifying the tasks are actually of high quality**
>
> - Thank you for raising this concern. We will revise the appendix to include several full task examples, and we would like to clarify that the full benchmark— including all questions, answers, and required MCP tools— is already provided in the supplementary zip file. In particular, the file `{mcpverse-main 2}/mcpverse/data/mcpverse_benchmark2.xlsx` contains the complete set of tasks used in our experiments, which can be directly inspected to assess their quality, diversity, and difficulty.
>
> - we do intend to open-source the entire benchmark and evaluation system facilitate reproducibility of our results, and we apologize for not stating this clearly in the current submission.
>
> > **The authors mention "After initial construction, every task undergoes a comprehensive review." --> what was this review, and how can we trust the quality?**
>
> - Thank you for your comment. As noted above, the complete benchmark is included in the supplementary materials and can be inspected directly to assess task quality.
>
> - The “comprehensive review” refers to an additional verification pass conducted by the authors after the initial construction of all tasks. In this stage, we manually reviewed every task and filtered out any items that did not satisfy the criteria specified in Section 3.2 (Curation Protocol).
>
> > **it's hard to tell whether the results are due to limitations of the API endpoint or the scaffold versus the models themselves**
>
> - We appreciate the reviewer’s concern. As stated in our Curation Protocol (Section 3.2), we only keep tasks for which a valid solution path exists within the selected set of MCPs, and discard any that do not satisfy this criterion. 2) Our evaluation system performs a preliminary check of all MCP server connections to ensure that tools are available and functioning as expected before running experiments.
>
> > The binary evaluation metric for success is a bit confusing -- is there a reason we didn't consider partial credit?
>
> - We thank the reviewer for this question. Our choice of a binary success metric follows common practice in recent tool-use benchmarks[1], where a task is counted as correct only if the final answer satisfies the evaluation predicate. While some prior work introduces partial credit by checking whether the tools used in the trajectory match a single annotated sequence, this is feasible only when the tool set is small and the “correct” tool choice is essentially unique at each step. In MCPVerse, the number and diversity of tools are much larger, and multiple tool sequences can be equally valid, making trajectory-based partial credit brittle and ill-defined.
>
> >  Typos
> - We will correct this in the revised manuscript.
>
> We appreciate the reviewer’s helpful feedback and hope that our clarifications address the concern and may lead to a reconsideration of the corresponding score. Thank you again for your time and insights!
>
> [1] Berkeley Function-Calling Leaderboard (BFCL) V4

---

### Meta-Review · Area_Chair_rzc7 · 2026-01-07

**Summary:**

The paper introduces a new benchmark for agentic tool use called MCPverse for evaluating the tool usage for reasoning agents. This benchmark includes 65 MCPs and 552 tools, with a variety of task types, complexities, and time sensitivities. The authors evaluate models from a variety of providers and share the results across three modes (Oracle, Standard, Max-Scale).

**Reviewer Concerns:**

- Lack of details and example tasks
- Key-policy conflict.
- Minor but impactful wording issues
- The task design is manual

**Reviewer Scores:**

they will remain the score

---

### Decision · Program_Chairs · 2026-01-26

Reject